# Evaluations of the Gap between Supervised and Reinforcement Lifelong Learning on Robotic Manipulation Tasks

**Fan Yang, Chao Yang, Huaping Liu**[*] **Fuchun Sun**
Department of Computer Science and Technology, Tsinghua University, China
Beijing National Research Center for Information Science and Technology, China
[*]Corresponding author: Huaping Liu (hpliu@tsinghua.edu.cn)

**Abstract:** Overcoming catastrophic forgetting is of great importance for deep learning and robotics. Recent lifelong learning research has great advances in supervised learning. However, little work focuses on reinforcement learning(RL). We focus on evaluating the performances of state-of-the-art lifelong learning algorithms on robotic reinforcement learning tasks. We mainly focus on the properties of overcoming catastrophic forgetting for these algorithms. We summarize the pros and cons for each category of lifelong learning algorithms when applied in RL scenarios. We propose a framework to modify supervised lifelong learning algorithms to be compatible with RL. We also develop a manipulation benchmark task set for our evaluations.

**Keywords:** Lifelong Learning, Reinforcement Learning, Manipulation

## 1 Introduction

Humans and animals continuously learn different skills and knowledge in their lifetime. It is natural for them to master a new skill without forgetting those learned before in most cases. However, neural networks suffer a lot from memorizing previously learned tasks. This phenomenon is called catastrophic forgetting [1, 2, 3]. Recent research has focused on this problem and proposed various solutions called lifelong learning or continual learning.

On the other hand, lifelong learning is of great importance for robotics. It provides possible solutions for developmental robotics [4]. Humans learn different skills as they grow up rather than completely rely on innate skills. It would also be difficult to pre-program all the required skills for advanced robots when setting up in the factory. Future robots will have to continuously master different tasks as humans do in their lifetime. The ability of lifelong learning will make robots fit better in various applications.

In this work, we focus on the lifelong learning problem for robotic manipulation tasks in the reinforcement learning(RL) domain, especially forgetting properties. Most of the recent lifelong learning research focuses on supervised learning. However, lifelong learning for RL is much more challenging than supervised learning. Not only because the training of RL itself is challenging and unstable[5], but in most RL tasks, policies are sensitive to forgetting. Forgetting could be considered as an attack in the action space of the agent. Recent work has demonstrated the vulnerability of RL agents to attack on action space[6]. Moreover, in RL tasks, an action at one step will influence states and actions followed up, which could aggravate the influence of forgetting. For example, in grasping tasks, slight misplacement of a fingertip is likely to break the balance of a grasp and lead to its failure. Though recent lifelong learning research has promising performances on supervised learning, little evidence has proven that they could directly be applied in robotic RL tasks. Research on how the state-of-the-art lifelong learning algorithms perform on robotic reinforcement learning tasks is urgently needed in the research community, which is the major motivation of our work.

In this paper, we establish a framework to study the possibilities to extend the existing supervised lifelong learning method to the reinforcement scenarios, demonstrating their effectiveness on a

5th Conference on Robot Learning (CoRL 2021), London, UK.

carefully-designed multi-task manipulation scenario. Our contributions can be summarized into three-folds[1]:

1. We evaluate Soft-Actor-Critic[7] with the state-of-the-art lifelong learning algorithms from four different categories using our benchmark. Forgetting performances and robustness to task orders are evaluated in experiments. We further analyze the possibility of applying different lifelong learning algorithms into RL based on our experiments.

2. We propose a manipulation benchmark for lifelong learning with ten different manipulation tasks. Policies trained on our benchmark could quickly converge.

3. We propose a framework which could modify traditional lifelong learning algorithms from supervised learning into reinforcement learning.

## 2 Related Works

### 2.1 Reinforcement Learning with Multiple Tasks

In addition to lifelong learning, meta-learning and multi-task learning both include multiple tasks in their problem formulations. They are similar domains but share plenty of differences. It is important to notice that in traditional lifelong learning formulations, **tasks are assigned to the agent sequentially**, which means that the agent will not be able to have access to the training data and environments after finishing training on this task. Furthermore, recent researches in lifelong learning focus on overcoming catastrophic forgetting and memorizing all the tasks learned. Though the abilities of transfer learning may also be discussed, they are not the core questions in this paper.

In the following sub-sections, we will discuss meta-learning and multi-task learning. Especially on the differences and similarities between these domains and lifelong learning.

#### 2.1.1 Meta-Learning

In meta-learning, research focuses on achieving fast adaptation based on agents' previous learning procedures. Specifically, [8] introduces an additional gradient descent procedure to update the network parameters for a novel task. [9] applies meta-learning to quickly adapt to a new dynamic model in a new environment, which leads to a faster generalization in a new environment. [10, 11] introduces a latent variable to represent the dynamic environment, and it could be used to help fast adaptation of robotic tasks.

#### 2.1.2 Multi-Task Learning

Multi-task learning is defined as solving multiple tasks at the same time. Multi-task learning includes lifelong learning but is not limited to it. Recent work on multi-task learning focuses on exploiting the similarities of tasks to accelerate the learning of a novel task. Specifically, [12, 13] focuses on how auxiliary tasks help improve the performance and learning speed of current tasks. Distral[14] introduces a condensed shared policy for faster learning of a new task. IMPALA[15] and its updated version PopArt-IMPALA[16] uses multiple agents to explore and train in different tasks at the same time. Note that in multi-task learning, tasks are not required to be learned sequentially, which is different from lifelong learning.

In summary, lifelong learning, meta-learning, and multi-task learning are mutually similar in the RL domain. However, they focus on different performances. In lifelong learning, tasks are trained sequentially and the performance of memorizing policies for previous tasks is highlighted.

### 2.2 Lifelong Learning in Reinforcement Learning

In terms of overcoming catastrophic forgetting, most lifelong learning algorithms only focus on solving supervised learning tasks, while work on RL has received little attention. There are some algorithms trying to solve RL tasks, but they are applied in image-based video games instead of robotic tasks, e.g., [17] can only be used in video games due to its generative image-based methods. [1] and [18] evaluates their methods only on Atari video games instead of robotic tasks. Some

---

[1]Details of the project can be seen on our website: https://lifelongreinforcementlearning.vercel.app/.

work on lifelong learning is applied in the robotics domain, but they choose preliminary tasks and they only provide little insight on how the state-of-the-art lifelong learning algorithms perform in robotic tasks. For example, [19] is implemented to real robots, but their method can be only used in locomotion tasks with a limited control space.

[20]evaluated lifelong learning on reinforcement tasks for robotics. It is the most related work in reinforcement lifelong learning. They modify the method mentioned in [15] to make it compatible with lifelong settings. Though having good performances, their method is only evaluated on up to three tasks in sequence. [21] presents an algorithm which demonstrates exceeding performance in lifelong learning and multi-task learning. However, it focuses on how previous learning experiences would help boost the learning of a new related task, while discusses little about catastrophic forgetting, which is the major concern in lifelong learning.

In conclusion, the lifelong learning problem, especially catastrophic forgetting, of reinforcement learning in robotics is poorly studied. An overall investigation about how the state-of-the-art lifelong learning methods perform on robotic reinforcement learning tasks is needed, which it the motivation of our work to modify traditional lifelong learning algorithms and compare their performance in RL.

## 3 Preliminaries

### 3.1 Markov Decision Process

We formulate our the problem as a Markov Decision Process (MDP). We use $s_t^k \in \mathbb{R}^{d^s}$ to denote the state of the agent at time step $t$ in task $k$. The action vector $a_t^k \in \mathbb{R}^{d^a}$ are sampled from the policy $\pi_\theta(a_t^k)$, where $\theta$ denotes the policy parameters. $p(s_{t+1}^k|s_t^k, a_t^k)$ denotes the probability of transforming from the state $s_t^k$ to $s_{t+1}^k$ by executing the action $a_t^k$. The agent will receives a reward $r(s_t^k, a_t^k, s_{t+1}^k)$ at each step. Our goal is to maximizing the sum of the discounted reward for all the tasks $\mathbb{E}_{\pi_\theta}[\sum_{k=1}^{K} \sum_{t=0}^{\infty} \gamma^t r(s_t^k, a_t^k, s_{t+1}^k)]$

### 3.2 Actor-Critic Framework

Actor-Critic-based algorithms compose a majority of RL algorithms. [7, 22, 23] all incorporate the framework. In traditional supervised learning problems, the loss for each trial can be computed by comparing the outputs with the ground-truth labels. However, in most reinforcement learning cases, such labels do not exist. Instead, in the Actor-Critic framework, a Critic is trained simultaneously with an Actor to evaluate the action the Actor chooses. The Critic loss is usually defined by the mean square Bellman error.

$$L(\phi) = \mathbb{E}[(Q_\phi(s_t^k, a_t^k) - (r_t^k + \gamma(1 - d_t^k) \max_{a_{t+1}^k} Q_\phi(s_{t+1}^k, a_{t+1}^k)))^2], \tag{1}$$

where $Q$ denotes the Critic network with parameters $\phi$, $d_t^k$ denotes whether the state $s_{t+1}^k$ is terminal state in $k$-th task. The Actor loss is defined by: $L(\theta) = -Q_\phi(s_t^k, \pi_\theta(s_t^k))$.

## 4 Reinforcement Lifelong Learning Framework

In this section, we propose the framework to modify traditional supervised lifelong learning algorithms to reinforcement learning. We purposefully design this framework to be as simple as possible. Our main contribution lies in the evaluation of lifelong learning algorithms. We want to minimize the influence of the additional network structure on the evaluation results.

Traditional lifelong learning algorithms cannot be directly applied to RL because of the following two reasons: (a) the loss in supervised learning is clearly defined, which is the error between predictions and labels. However, it is not straightforward in RL. A critic network is needed for the evaluation of the policy network. (b) It would be hard to indicate which task the robot is performing, while in supervised learning, e.g., in image classification, the network usually does not need to know the task ID since the images themselves have sufficient information to indicate which task they are within. In order to solve these two problems, we modify the traditional lifelong learning algorithms in the following ways.

### 4.1 Actor-Critic Framework in the Lifelong Learning Algorithms

We adopt the Actor-Critic algorithm into our lifelong learning framework to have a parametric evaluation of the actor policy. we replace the loss (usually MSE or cross entropy) in traditional lifelong learning algorithms with critic loss. In addition, since in real-world applications, critics are usually discarded and only the actor (policy network) is used, we only consider the continual learning properties of the actor while we use traditional deep RL algorithms to learn task-specific critics. However, it is also straightforward to expand the lifelong learning algorithms for the critics.

### 4.2 Task-Specific Sub-Network

The observation spaces in RL tasks could be identical. In such cases, the policy network will have to distinguish different tasks by task IDs. It would be challenging if the task ID is directly input to the network for learning. Instead, we add task-specific layers into the policy network if lifelong learning algorithms themselves do not include such task-specific parts. Formally, given $k$ tasks, the parameters of the policy network $\pi_\theta$ can be decomposed into common parts and specific parts: $\theta = \{\theta_c\} \cup_{i=1}^{K} \{\theta_s^i\}$. For task $i$, the action $a_t^i$ can be given by: $a_t^i = \pi_\theta(a_t^i) = \pi_{\theta_s^i}(\pi_{\theta_c}(s_t^i))$. Different sub-networks are chosen for different tasks according to task IDs. In our experiments, the task-specific sub-networks are only a single-layer affine transformation. If sub-networks with deeper structures are implemented, the shared parts could be simply an identity transformation while the policy totally relies on the sub-network, which is basically training a policy network for each task.

Details are shown in the appendix.

## 5 Lifelong Manipulation Benchmarks

In this section, we introduce the lifelong manipulation benchmarks. Our benchmarks adapt from ROBEL D'Claw[24], a mujoco-based robotic manipulation platform. We modify the tasks and get ten different manipulation tasks. The benchmarks consist of a robotic hand with three independent fingers. Each finger has three degrees of freedom. In our task, the robotic hand needs to turn the valve below 180 degrees. The observation space of the benchmarks consists of joint angles, joint velocities, and the valve angle. The action space of the robot consists of robot joint angles. The reward function is designed based on robot configurations and the valve position to ensure a safe manipulation. Each task only differs in the valve shape. Though each task seems to be similar, a policy trained on a single task cannot achieve high rewards on the other tasks. All the tasks are designed to have dense rewards for a faster learning procedure. Note that the absolute reward value itself does not have much meaning. A higher or a lower reward only indicates a better or worse performance. Details of the reward function can be seen in [24].

Though Meta-World[25] provides a multi-task benchmark and it is likely to be applied in lifelong learning, Their tasks are far more challenging than ours. Tasks in Meta-World could be difficult in lifelong learning settings. Our benchmarks are used for different levels of evaluations.

One of the important challenges in reinforcement lifelong learning lies in the training time. RL policies are notably hard to train. This problem will be aggravated in the lifelong learning scenarios since each task should be trained one after another by definition. In our benchmark, each task is stable and easy to train. In our experiments, each policy can converge after 80,000 steps of updates using SAC. The details of the benchmark could be seen in Fig. 1.

## 6 Experiments

### 6.1 Evaluation Metrics

We will mainly focus on the forgetting of lifelong learning. We evaluate each algorithm using the average performance. It is slightly modified from the average accuracy, a standard lifelong learning evaluation metric used in [26][27] and [28].:

**Average Performance**: Let $p_{i,j}$ be the cumulative reward of the task $j$ after the model finishes learning task $i$. Then the average performance $P$ is defined as: $P = \frac{1}{K} \sum_{j=1}^{K} p_{K,j}$, where $K$ denotes the total number of tasks. When evaluating, since we use the cumulative reward to evaluate

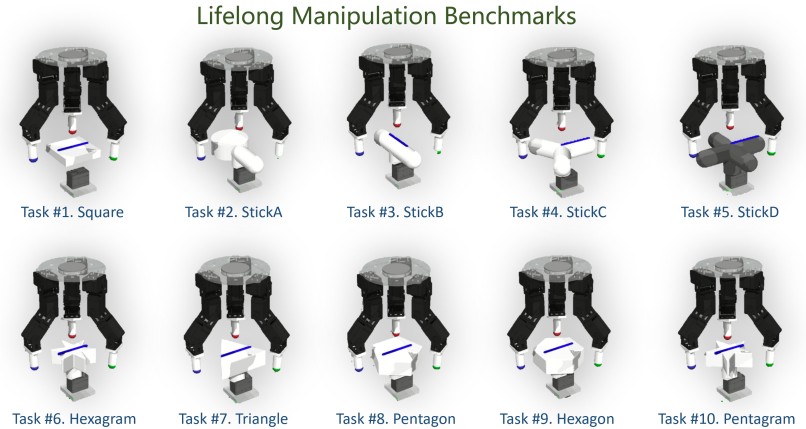

Figure 1: 10 manipulation tasks for lifelong manipulation benchmarks

policies, the time steps for each episode are set to be constant for fairness, no matter whether the task has already converged.

## 6.2 Taxonomy of Lifelong Learning

Recent lifelong learning algorithms can be classified into the following four categories:

### 6.2.1 Regularization-Based Methods

Regularization-based methods add an additional regularization term in the loss function to ensure the learned network in the new task will not forget important information for the previous tasks. The most intuitive method would add an MSE loss to minimize the parameter changes between the current task and previous tasks. Instead, [1] proposed Elastic Weight Consolidation(EWC). It incorporates the Fisher information matrix to compute the importance of network parameters and tries to keep the important ones unchanged. The most important advantage of regularization-based methods is that the model size will not change when the number of tasks increases.

### 6.2.2 Memory-Based Methods

Memory-based methods would use a tiny replay memory to store several samples from previously learned tasks and replay when learning new tasks. The key idea of memory-based methods is trying to transform lifelong learning problems into classic deep learning tasks. Episodic Replay(ER)[28] combines samples from previous tasks with current tasks when computing the loss for the current task. [29] is an updated version of [28], which adds weight to learn the importance of the current task and previous task.

### 6.2.3 Gradient-Constraint Methods

Gradient-constraint methods modify the gradient to ensure that the update for the current task would have minimal influence on previous tasks. Gradient Episodic Memory(GEM)[26] projects the gradient into spaces to ensure the update for the current task will not decrease the performance of previous tasks. [27] proposed A-GEM and optimize the computation efficiency of GEM. Orthogonal Gradient Descent(OGD)[30] believes moving orthogonally to the gradients of previous tasks would have the smallest influence on them. Therefore, it projects the gradients for the current tasks into the orthogonal space of previous tasks. Continual Learning in Orthogonal Subspaces(CLOS)[31] designs a framework to ensure that the gradients are orthogonal by nature. It combines gradient-constraint methods and memory-based methods for a better performance.

### 6.2.4 Expansion-Based Methods

Expansion-based methods add a new sub-module when learning a new task. Thus, the size of the model increases with the learning procedure. Additive Parameter Decomposition(APD)[32] includes a shared parameter space with task-specific sub-networks. Learning a new task would add a new sub-network and the algorithm would also constrain the changes in the shared parameters.

In order to have an overall comparison of lifelong learning algorithms on RL, we choose one or two algorithms from the categories mentioned above. **Oracle** means that each task is trained with an individual policy. Oracle does not consider the lifelong learning problem. It is used to indicate the best possible performance each algorithm can achieve. **Random** means using a random agent in the benchmark tasks. It is used to indicate the worst possible performance for each algorithm. **SAC**[7] means using the original Soft-Actor-Critic algorithm trained continuously from one task to another. However, for the fairness of comparison, we also modify the actor of SAC into shared parts and task-specific parts. It is designed to demonstrate the performance of the state-of-the-art RL algorithms without lifelong learning. **EWC**[1] and **L2** are representatives of regularization-based methods. L2 only introduces an MSE loss for the network differences between the current actor parameters and previous task parameters, as is mentioned in Sec. 6.2.1. L2 is the most intuitive lifelong learning algorithm. **AGEM**[27] and **GEM**[26] and **CLOS**[31] are chosen as representatives for gradient-constraint methods. **ER**[28] is chosen for memory-based methods. **APD**[32] is chosen for expansion-based methods. In addition, **APD** and **CLOS** are the state-of-the-art ones to demonstrate the performance of the frontier research on lifelong learning.

## 6.3 Results

We first show the performances on each task after finishing learning the last task. Each algorithm is trained and updated with nearly the same steps for a fair comparison. The environments for the previous tasks are not allowed to be re-accessed after finishing the training. All the algorithms are trained with the same task order. The results are shown in Fig. 2. The average performance is shown in Table 1.

In Fig. 2, SAC only achieves a high reward on the last task. It indicates that traditional RL cannot solve the lifelong tasks designed in our benchmark tasks. Previous tasks still have a slightly higher performance than a random agent, which suggests each task is not totally independent from others. L2 achieves worse results than SAC. It indicates that improper constraints of lifelong learning, e.g., regularization, are not only detrimental to memorizing previous tasks but will impede the learning of the current task. In comparison, EWC achieves better results. However, it has a much worse forgetting performance compared to its applications in supervised learning. It could result from the sensitiveness of manipulation tasks mentioned in Section 1. GEM has better performances than AGEM. Compared to AGEM, GEM samples more data from previous tasks when updating. It could lead to a more conservative update, which is important in the sensitive RL environment. Though simple and intuitive, ER achieves a good performance. It is worth noting that in RL, we do not need to design additional replay memories since RL algorithms will need to use replay memories themselves. CLOS includes memory-based methods into gradient-based methods by replaying previous tasks when training. It achieves better performances than gradient-based ones, which suggests that memory-based methods could help improve the performance of other lifelong algorithms. Though APD achieves a good result, expansion-based methods will add a sub-module when learning a new task in practice. The size of the added sub-module could be as large as training a new policy from scratch, which degrades the effect of lifelong learning.

Table 1: The evaluation performance of all the lifelong learning algorithm for manipulation benchmarks.

| Algorithm | Oracle | Random | SAC | L2 | EWC | AGEM | GEM | ER | CLOS | APD |
|-----------|--------|--------|-----|----|-----|------|-----|----|------|-----|
| Mean | 1709.44 | -810.432 | -535.57 | -618.87 | -249.73 | -286.44 | 549.52 | 867.21 | 1543.90 | 1379.65 |
| Std. | 131.97 | 692.87 | 64.01 | 98.11 | 824.50 | 736.11 | 929.91 | 586.73 | 259.49 | 171.25 |

To better demonstrate the forgetting details of each algorithm, we provide figures of cumulative rewards during the training procedures. The results are shown in Fig. 3. The x-axis denotes training steps. The y-axis denotes cumulative rewards. Each column shows performances of a policy for all

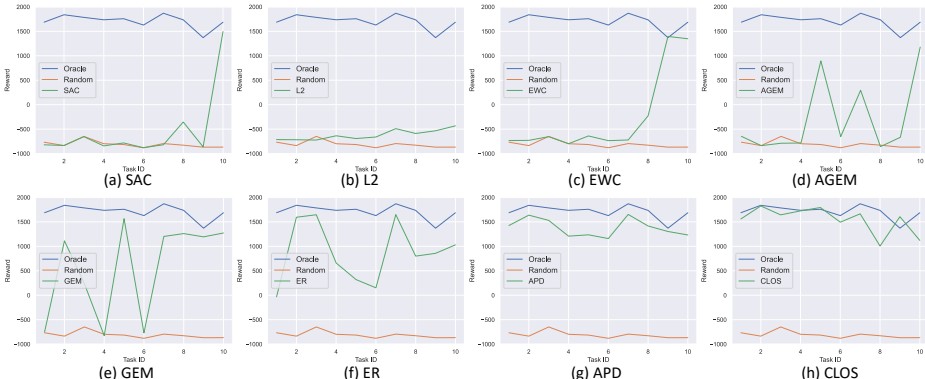

Figure 2: Performance of lifelong learning methods with the RL framework mentioned in Sec. 4. The x-axis denotes task IDs. The y-axis denotes cumulative rewards. Each reinforcement lifelong learning algorithm consists of a lifelong learning algorithm and SAC as the RL algorithm. The figures show the performances of the policy on each task after finishing learning the last one. Oracle and random agent is included in each figure for comparison.

the tasks, from before training a certain task to after training the last task. It is used to demonstrate the influence of training other tasks on the current task.

In Fig. 3, the performance for each task in SAC only peaks when training it, which means it quickly forgets the policy for previous tasks. The rewards in ER, APD, CLOS will keep at a high level even after training a certain task. CLOS combines several methods together and has the best performance among the algorithms mentioned above. However, the training of CLOS is less stable than APD and ER. Specifically, the performance will vary a lot during the training procedure. One has to save several policies and search for the best one when using CLOS.

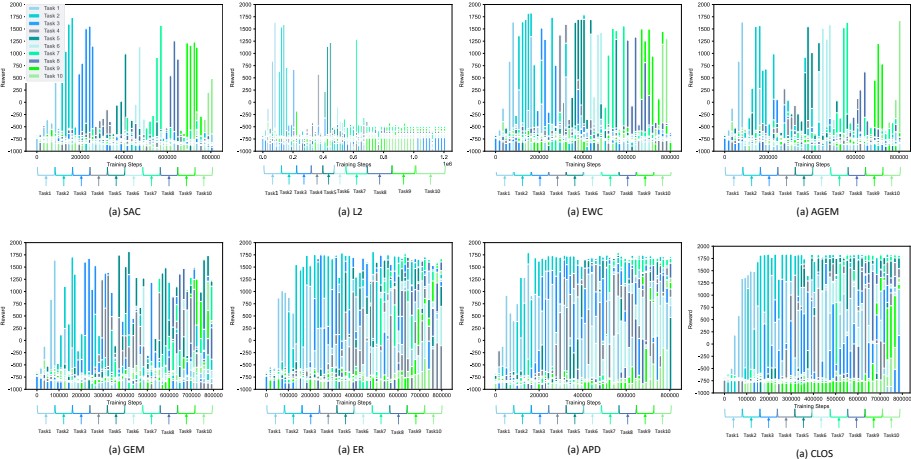

Figure 3: Performance changes of lifelong learning methods along with the RL training procedure. Different lifelong learning algorithms are trained with the same task order. Each column demonstrates the cumulative reward of the policy from before training on a task to finishing training the last task.

A reliable lifelong learning algorithm would not be influenced by the task orders. We train each algorithm in three different task orders and evaluate the policies on all the ten tasks after finishing training the last task. Results are shown in Fig. 4.

Task order experiments in Fig. 4 demonstrate that the algorithms with satisfying performance in the previous experiments are usually also robust to order changes. APD demonstrates the best robustness

to task orders, since expanding the network makes it similar to training an independent policy for each task.

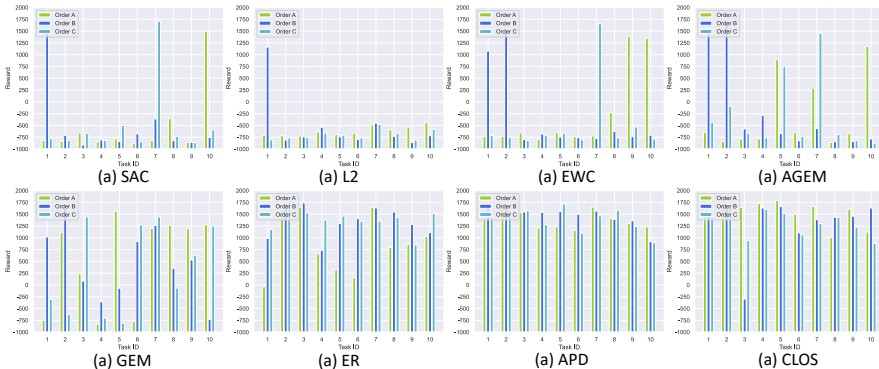

Figure 4: Performances of lifelong learning algorithms trained with different task orders. Performances are evaluated after training the last task. Columns with the same color indicate experiments within the same task order.

## 7 Discussion

In this section, we conclude several concise wake-away conclusions of our evaluations.

Firstly, **RL tasks would intensify the weakness of lifelong learning algorithms.** EWC, GEM, and AGEM, which are notably effective in supervised lifelong learning, have a poor performance in RL. Even some could have a good result on supervised learning, the performance could be dramatically degraded in RL. Specifically, **lifelong learning in robotic manipulation tasks is far more sensitive than that in supervised learning.** In RL settings, the gaps between different lifelong learning methods are greater than that in supervised learning.

Secondly, **expansion-based methods and memory-based methods have a good performance on lifelong RL methods.** Future work could consider incorporating these two methods as a part of its algorithm for a better performance. Regularization-based methods and gradient-based methods still suffer in RL tasks. But on the other hand, the model size is limited in these methods, which has great potential in real robot applications.

Thirdly, **existing lifelong learning methods requires memorizing lots of data, which makes it impractical in robotic applications.** They require memorizing samples or models from previous tasks. However, for applications on real robots, it will not be possible for the robot to remember too many samples or will not allow a substantial increase in the model size. Existing lifelong learning methods still do not have a perfect solution to it.

We also point out that **a more comprehensive evaluation metrics in reinforcement lifelong learning is needed.** Unlike supervised learning, even if the cumulative reward of a task could decrease greatly, it does not really mean the policy forgets greatly about the task. Since RL tasks could be highly sensitive and the policy might not be robust, a slight forgetting could lead to a complete failure of the task.

## 8 Conclusion

In this paper, we focus on the problem of how the state-of-the-art lifelong learning algorithms perform on RL manipulation tasks. We design a framework to modify traditional lifelong learning algorithms used in supervised learning into the ones in RL. We also develop a manipulation benchmark for evaluation. We test seven lifelong learning algorithms with three baselines on our benchmark. According to our experiments, we point out the sensitiveness challenge of lifelong RL tasks compared to lifelong supervised learning tasks. We also summarize the potentials of applying different kinds of lifelong learning tasks in RL.

## Acknowledgements

This work was supported in part by the National Natural Science Foundation of China under Grant 62120106005, and the Joint Fund of Science & Technology Department of Liaoning Province and State Key Laboratory of Robotics, China (2020-KF-22-06).

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
