# OpenReview forum: "Evaluations of the Gap between Supervised and Reinforcement Lifelong Learning on Robotic Manipulation Tasks"
_robot-learning.org/CoRL/2021/Conference — CoRL2021 Poster_

### Official Review · Reviewer_x2T2 · 2021-07-21

**Originality:** Good
**Technical Quality:** Very Good
**Clarity Of Presentation:** Fair
**Impact:** 3

**Recommendation:**

Weak Reject: I recommend rejecting the paper, but will not argue for my recommendation if the majority of other reviewers have a different opinion.

**Summary:**

The main contribution of this paper are summarized as follows:
The manuscript proposes a novel framework which could modify traditional lifelong learning algorithms in supervised learning into reinforcement learning.
The manuscript proposes a manipulation benchmark for lifelong learning with ten different manipulation tasks.
The manuscript evaluates Soft-Actor-Critic with the state-of-the-art lifelong learning algorithms from four different categories using proposed benchmark, and evaluates forgetting performances and robustness to task orders in experiments. In the end, the manuscript analyzes the potential of different lifelong learning algorithms applied in RL based on experiments.


**Issues:**

1) The novel reinforcement lifelong learning framework in this manuscript is similar with overcoming catastrophic forgetting algorithm framework in "Learning without Forgetting". It would be better to provide more details about difference between novel reinforcement lifelong learning framework and the framework in paper "Learning without Forgetting".
2) It would be better to provide the comparative experiment with more details for novel reinforcement lifelong learning framework and state-of-the-art baselines. Current tables and figures in manuscript could support the advantage of manipulation benchmark for lifelong learning rather than the advantage of performance for proposed novel reinforcement lifelong learning framework.
3) It would be better to unify proper noun. For example, the manuscript states that " Reinforcement Lifelong Learning " in title, and states that " Lifelong Reinforcement Learning " in content.
4) The manuscript states that "Though recent lifelong learning research has promising performances on supervised learning, little evidence has proven that they could directly be applied in robotic RL tasks." in introduction. However, lifelong reinforcement learning has been studied for several years. For example, "Online Multi-Task Learning for Policy Gradient Methods". It would be better to pay more attention in investigation.

**Reviewer Expertise:**

Very good: Comprehensive knowledge of the area

**Strengths And Weaknesses:**

The main advantages of this manuscript are two aspects. First, this manuscript proposes a novel reinforcement lifelong learning framework in order to make reinforcement learning algorithm owns the ability of overcoming catastrophic forgetting for previously learned tasks, and owns the ability of speeding up the convergence rate for new task. Second, this manuscript proposes a manipulation benchmark for lifelong learning with ten different manipulation tasks.
However, there are some technical problems left at current version, some of which are given as follows.
1) The manuscript states that "Though recent lifelong learning research has promising performances on supervised learning, little evidence has proven that they could directly be applied in robotic RL tasks." in introduction. However, lifelong reinforcement learning has been studied for several years. For example, "Online Multi-Task Learning for Policy Gradient Methods". It would be better to pay more attention in investigation.
2) The novel reinforcement lifelong learning framework in this manuscript is similar with overcoming catastrophic forgetting algorithm framework in "Learning without Forgetting". It would be better to provide more details about difference between novel reinforcement lifelong learning framework and the framework in paper "Learning without Forgetting".
3) It would be better to provide the comparative experiment with more details for novel reinforcement lifelong learning framework and state-of-the-art baselines. Current tables and figures in manuscript could support the advantage of manipulation benchmark for lifelong learning rather than the advantage of performance for proposed novel reinforcement lifelong learning framework.
4) It would be better to unify proper noun. For example, the manuscript states that " Reinforcement Lifelong Learning " in title, and states that " Lifelong Reinforcement Learning " in content.

**Summary Of Recommendation:**

The related investigation was not enough. And current tables and figures in manuscript could not support the advantage of performance for proposed novel reinforcement lifelong learning framework.

---

> ### Author Response · Authors · 2021-08-29
> **Our main contribution lies in evaluation of the sota lifelong algorithm on robotic RL tasks, which is important but has received little attention.**
>
> Thank you for your constructive comments. We are glad that our lifelong learning benchmark and evaluations are recognized to some extent. We have highlighted our revised part in cyan in our updated version. Here are our replies to your helpful suggestions.
> 1. For the lifelong learning framework issue, we thank you for providing suggestions on the comparison. We think there is a little bit of misunderstanding. Our main contribution would be the evaluations of the sota lifelong learning algorithms on robotic reinforcement learning tasks since this question is poorly studied and of great importance in the robotics community. We purposefully design the framework to be as simple as possible, so that the framework will have less influence on the evaluation results. However, we realize that the presentation of our paper has to be improved so that our main contribution could be highlighted and misunderstanding could be prevented. Therefore, in our revised version, we shortened the framework section and added sentences to highlight our main contribution.
> 2. Ideal lifelong learning algorithms would both overcome catastrophic learning and quickly adapt to a new task as human beings do. However, the algorithms we chose consider little about these transferring properties in their original paper. Therefore, we mainly focus on catastrophic forgetting in our work. Catastrophic forgetting is also a major concern in recent lifelong learning research. It is important to consider transferring in lifelong learning. We do not think the lifelong learning algorithm we chose for evaluations will have promising results. Instead, multi-task learning and meta-learning may have a good performance on this issue, which we will include in our future work on comparing transferring abilities.
> 3. We did not compare our framework to the sota baselines for the following reasons. Firstly, only a  few baselines could be found on this topic, most do not even share the same problem formulations. Secondly, our main contributions lie in evaluations of sota lifelong learning algorithms on RL tasks. We agree it is important and we will consider it in our future work due to the space limitations in this work.
> 4. Thank you for reminding us. "Online Multi-Task Learning for Policy Gradient Methods" does propose methods which could be effective in lifelong learning area. We have added it to our revised version. Though related, this paper hardly focuses on catastrophic forgetting, which is the major concern in lifelong learning and our paper. It pays more attention to how previous tasks would help speed up the learning of a novel task. We realize that it would be a little bit inappropriate to emphasize that little work has been done in reinforcement lifelong learning. However,  the reinforcement lifelong learning area does require more attention and it's poorly studied. We realize that we missed this related work because we only focused on lifelong learning, especially catastrophic forgetting, when reviewing the literature. Multi-task learning and even meta-learning are closely related to lifelong learning. Thus we add these parts in our revised version.
> 5. Thank you for the suggestions on the proper noun. We have updated it

---

### Official Review · Reviewer_FpJK · 2021-07-23

**Originality:** Good
**Technical Quality:** Fair
**Clarity Of Presentation:** Good
**Impact:** 3

**Recommendation:**

Weak Reject: I recommend rejecting the paper, but will not argue for my recommendation if the majority of other reviewers have a different opinion.

**Summary:**

This work provides a framework for evaluating various life long learning techniques in a robotic reinforcement learning context. Several approaches (regularization, memory, gradient, and expansion based) are compared. Each approach must learn a set of 10 knob turning tasks in sequence, and performance on all 10 is recorded afterwards.   The memory and expansion based approaches outperform the regularization and gradient based approaches at the cost of requiring replay buffer storage of past episodes or a growing policy size.

**Issues:**


- Table 1: Could you elaborate on what a negative score means? Is the knob spinning in the opposite direction?

**Reviewer Expertise:**

Good: General knowledge of the area

**Strengths And Weaknesses:**

Strengths:
The evaluation setup from this paper is well motivated and implemented.  It captures a temporal nature of tasks changing over time, different from a multi-task setup where multiple tasks are learned concurrently.

Weaknesses:
- This work doesn't propose a novel method, only a benchmark and comparison of existing methods.

Perhaps another interesting dimension in addition to the forgetting of previous tasks would be number of rollouts required to learn a new task to some threshold success rate. For example, one would hope a capable life long learning agent with more past experiences would be able to learn a new task more quickly than an agent with less past experiences to leverage.

**Summary Of Recommendation:**

The framework for a life long learning benchmark is a great idea.  That being said, the benchmark articulated in this work leaves a good bit to be desired. It would be interesting if other dimensions beyond catastrophic forgetting where to be discussed.  For example, looking at how number of rollouts needed to learn a new task changes overtime.

It is also a bit unclear to me how well a set of 10 very similar tasks captures the requirements needed for a life long learning system.

---

> ### Author Response · Authors · 2021-08-29
> **We mainly focus on overcoming catastrophic forgetting in this paper and the Ten benchmark tasks are designed similar but essentially different.**
>
> Thank you for your constructive suggestions. We have highlighted our revised part in cyan in our updated version. Here are our replies:
> 1. We appreciate your suggestions on evaluating the adaptability to a new task using past learning experiences. However, due to space limitations, we only consider catastrophic forgetting in this paper, which is also the major concern in the recent lifelong learning research. Many recent researches on lifelong learning only discuss about overcoming catastrophic forgetting in their paper, e.g., the EWC paper. Moreover, the algorithms we chose to evaluate hardly discuss the adaptation and transferring property in their original paper. It would be hard for these algorithms to have a satisfying performance on transferring properties. We will focus more on transferring properties in our future work. And it may include further work about meta-learning and multi-task learning, which discuss more about the transferring properties. We have revised our paper made it clear that we only focus on the forgetting properties.
> 2. Though ten tasks are similar to each other, we have verified that the policy trained on a specific task is not possible to solve other tasks by our preliminary experiments. Ten tasks in our benchmark are designed similar to each other. Ideally, each task should be distinct from others and some are related to each other to evaluate different properties. Meta-World has proposed a benchmark like this. However, we think the sota lifelong learning algorithms are not likely to solve such challenging tasks. Therefore, we design such benchmark though simple but effective in evaluations.
> 3.  The negative score does not have a specific meaning, a lower score only means a worse performance, which is usually not touching the valve. The reward score is exactly the same as that in ROBEL: RObotics BEnchmarks for Learning with low-cost robots.

---

### Official Review · Reviewer_R49v · 2021-07-24

**Originality:** Good
**Technical Quality:** Good
**Clarity Of Presentation:** Good
**Impact:** 4

**Recommendation:**

Strong Accept: I recommend accepting the paper and will argue for my recommendation even if other reviewers hold a different opinion.

**Summary:**

This paper evaluates 7 lifelong learning methods from supervised learning for reinforcement learning. It uses a credible, minimal lifelong learning framework based on SAC and a bespoke manipulation benchmark, based on ROBEL D'Claw.

**Issues:**

See comments above.

Detailed comments:
Line 95: I don't think it's ever possible for a method (which works well) to be "too simple and straightforward". The best method is no method at all! Please reconsider your phrasing here.

Figure 3: Please provide axis labels!

Table 1: Please have a clear heading for this table, and consistent use of rows/columns.

**Reviewer Expertise:**

Excellent: Expert knowledge on the topic of the paper

**Strengths And Weaknesses:**

Strengths
---
* Presents important experiments which establish a baseline for lifelong RL research
* High coverage of different LL methods and techniques
* Considers important LL questions, such as task ordering
* Simple paper which seeks to do one thing well

Weaknesses
---
* Editing and writing is sometimes weak; many sentences are awkward, grammatically incorrect, or not very idiomatic; missing labels on plots; awkward use of whitespace (use the ~ for spacing references, i.e. `Fig.~\ref{fig:my_figure}`!)
* I would like to see some more concise take-aways in the discussion section. As it is, it is quite dense.

**Summary Of Recommendation:**

I think evaluative and empirical work is very neglected in the field, and I would love to see more papers like this in the future.

I support accepting this paper because it contains important data and analysis for authors and reviewers working in lifelong reinforcement learning. The authors are correct that there are precious few resources with any information on the performance of popular LL methods on reinforcement learning, and especially few which do so using robotics challenges. This paper, in its best form (see my comments below and above) would be an asset for those studying LLRL for robotics.

I rated this paper weak accept because I believe it is held back by a few presentation issues (see my comments above). I think this paper could use a lot of attention in the discussion section, to provide readers with a few very clear take-aways from the analysis (and perhaps a summary table, if appropriate). Papers like this have the most impact as a quick reference for future researchers. Additionally, a general once-over on presentation, typesetting, grammar, and style would do wonders for the digestability of this work.

---

### Meta-Review · Area_Chair_y5qc · 2021-08-13

**Recommendation:** Accept (Poster)
**Confidence:** 4

**Metareview:**

**Updated meta-review after rebuttal**
After careful consideration of the reviews/rebuttals and paper revisions, I recommend accept.

This paper is focussed on an empirical evaluation of continual learning approaches applied to RL (continual learning approaches are typically proposed for the supervised learning setting, and not for the RL setting, so we have little insight on how these approaches would fare in the RL setup.) I agree with the reviewer that states that not proposing a novel method is not a good reason for reject (which is the major concern of the other 2 reviewers). The manuscript initially did have some issues on properly portraying the scope of the presented evaluation, but that was fixed.

**Initial meta-review**
**Summary**:
This paper benchmarks lifelong learning algorithms in the reinforcement learning on manipulation tasks. Current SOTA continual learning algorithms are adapted to the reinforcement learning setting, and then evaluated on a sequence of tasks.

**Strengths**:
Lifelong learning is an important problem of robotics, and this manuscript provides important insights on how current lifelong learning algorithms that were mostly developed for supervised learning problems perform in RL settings.

A good coverage and evaluation of existing methods

**Weaknesses**:

The reviewers point out that the metrics evaluated in this work are limited to measuring forgetting. However, lifelong learning is also about understanding how quickly a task can be (re-)learned given previously learned representations. The authors should consider adding such metrics to their evaluation.

Clarity and crisp take aways are lacking in a few places. The authors should carefully discuss and analyze the results, and improve the clarity of the manuscript

There is prior work on lifelong reinforcement learning (for simulated robotic tasks), and the authors should discuss such work. Given that this benchmark is also done in simulation, the authors should maybe not put too much emphasis on “little prior work for robotic lifelong RL exist” , but put more emphasis on “benchmarking of lifelong RL algorithms has received little attention”

---

> ### Author Response · Authors · 2021-08-29
> **Our work focuses on evaluations of catastrophic forgetting of lifelong learning in robotic RL tasks**
>
> Our work focuses on evaluations of catastrophic forgetting of lifelong learning in robotic RL tasks
> Thank you for your constructive suggestions, we are glad to see that our work on evaluation of lifelong learning is recognized. We have updated our manuscript according to the comments. Updated parts are shown in cyan in revised version. Here are our replies to some suggestions.
> 1. We appreciate your suggestions on adding new metrics into our evaluations, e.g.,  the adaptability to a new task using past learning experiences. However, due to space limitations, we only consider catastrophic forgetting in this paper, which is also the major concern in the recent lifelong learning research. Many recent researches on lifelong learning only discuss about overcoming catastrophic forgetting in their paper, e.g., the EWC paper. Moreover, the algorithms we chose to evaluate hardly discuss the adaptation and transferring property in their original paper. It would be hard for these algorithms to have a satisfying performance on transferring properties. We will focus more on transferring properties in our future work. And it may include further work about meta-learning and multi-task learning, which discuss more about the transferring properties. We have revised our paper made it clear that we only focus on the forgetting properties.
> 2.  Thanks for pointing out the clarity problems, we have updated them in our final version. We highlight our take-away conclusions and reorganize our manuscript.
> 3.  Thank you for reminding us. "Online Multi-Task Learning for Policy Gradient Methods" does propose methods which could be effective in lifelong learning criterion. We have added it to our revised version. Though related, it hardly focuses on catastrophic forgetting, which is the major concern in lifelong learning. It pays more attention to how previous tasks would help speed up the learning of a novel task. We realize that it would be a little bit inappropriate to emphasize that little work has been done in reinforcement lifelong learning. However,  the reinforcement lifelong learning area does require more attention and it's poorly studied. We realize that we missed this related work because we only focused on lifelong learning, especially catastrophic forgetting, when reviewing the literature. Multi-task learning and even meta-learning are closely related to lifelong learning. Thus we add these parts in our revised version.

---

### Decision · Program_Chairs · 2021-09-13

**Decision:**

Accept (Poster)

**Comment:**

**Updated meta-review after rebuttal**
After careful consideration of the reviews/rebuttals and paper revisions, I recommend accept.

This paper is focussed on an empirical evaluation of continual learning approaches applied to RL (continual learning approaches are typically proposed for the supervised learning setting, and not for the RL setting, so we have little insight on how these approaches would fare in the RL setup.) I agree with the reviewer that states that not proposing a novel method is not a good reason for reject (which is the major concern of the other 2 reviewers). The manuscript initially did have some issues on properly portraying the scope of the presented evaluation, but that was fixed.

**Initial meta-review**
**Summary**:
This paper benchmarks lifelong learning algorithms in the reinforcement learning on manipulation tasks. Current SOTA continual learning algorithms are adapted to the reinforcement learning setting, and then evaluated on a sequence of tasks.

**Strengths**:
Lifelong learning is an important problem of robotics, and this manuscript provides important insights on how current lifelong learning algorithms that were mostly developed for supervised learning problems perform in RL settings.

A good coverage and evaluation of existing methods

**Weaknesses**:

The reviewers point out that the metrics evaluated in this work are limited to measuring forgetting. However, lifelong learning is also about understanding how quickly a task can be (re-)learned given previously learned representations. The authors should consider adding such metrics to their evaluation.

Clarity and crisp take aways are lacking in a few places. The authors should carefully discuss and analyze the results, and improve the clarity of the manuscript

There is prior work on lifelong reinforcement learning (for simulated robotic tasks), and the authors should discuss such work. Given that this benchmark is also done in simulation, the authors should maybe not put too much emphasis on “little prior work for robotic lifelong RL exist” , but put more emphasis on “benchmarking of lifelong RL algorithms has received little attention”